# *Aristolochia trilobata*: Identification of the Anti-Inflammatory and Antinociceptive Effects

**DOI:** 10.3390/biomedicines8050111

**Published:** 2020-05-06

**Authors:** Dayana da Costa Salomé, Natália de Morais Cordeiro, Tayná Sequeira Valério, Darlisson de Alexandria Santos, Péricles Barreto Alves, Celuta Sales Alviano, Daniela Sales Alviano Moreno, Patricia Dias Fernandes

**Affiliations:** 1Laboratório de Farmacologia da Dor e da Inflamação, Instituto de Ciências Biomédicas, Universidade Federal do Rio de Janeiro, Rio de Janeiro 21941-902, Brazil; daycsalome@gmail.com (D.d.C.S.); natmoraiss@gmail.com (N.d.M.C.); tayna.sequeira@gmail.com (T.S.V.); 2Departamento de Química, Universidade Federal de Sergipe, Sergipe 49100-000, Brazil; darlisson@unifesspa.edu.br (D.d.A.S.); pericles@ufs.br (P.B.A.); 3Instituto de Ciências Exatas, Faculdade de Química, Universidade Federal do Sul e Sudeste do Pará, Marabá 68507-590, Brazil; 4Laboratório de Superfície de Fungos, Instituto de Microbiologia Professor Paulo de Góes, Universidade Federal do Rio de Janeiro, Rio de Janeiro 21941-902, Brazil; alviano@micro.ufrj.br (C.S.A.); danialviano@micro.ufrj.br (D.S.A.M.)

**Keywords:** *Aristolochia trilobata*, sulcatyl acetate, antinociceptive effect, anti-inflammatory activity

## Abstract

*Aristolochia trilobata*, popularly known as “mil-homens,” is widely used for treatment of stomach aches, colic, asthma, pulmonary diseases, diabetes, and skin affection. We evaluated the antinociceptive and anti-inflammatory activities of the essential oil (EO) and the main constituent, 6-methyl-5-hepten-2-yl acetate (sulcatyl acetate, SA). EO and SA (1, 10, and 100 mg/kg, p.o.) were evaluated using chemical (formalin-induced licking) and thermal (hot-plate) models of nociception or inflammation (carrageenan-induced cell migration into the subcutaneous air pouch, SAP). The mechanism of antinociceptive activity was evaluated using opioid, cholinergic receptor antagonists (naloxone and atropine), or nitric oxide synthase inhibitor (L-NAME). EO and SA presented a central antinociceptive effect (the hot-plate model). In formalin-induced licking response, higher doses of EO and SA also reduced 1st and 2nd phases. None of the antagonists and enzyme inhibitor reversed antinociceptive effects. EO and SA reduced the leukocyte migration into the SAP, and the cytokines tumor necrosis factor and interleukin-1 (TNF-α and IL-1β, respectively) produced in the exudate. Our results are indicative that EO and SA present peripheral and central antinociceptive and anti-inflammatory effects.

## 1. Introduction

Inflammation and pain continue to be major problems in individuals. Both of them are a normal response of the body against invasion and/or damage. Inflammation can develop in response to an invasion by a microorganism or by physical damage and is a critical protective action to injury or infection. This phenomenon presents the five cardinal signs (i.e., redness, heat, swelling, pain, and loss of function) [1]. Pain, one sign of inflammation, is an international health problem, affecting about one in five individuals. Both situations affect almost 20% of worldwide population. Drugs used to treat the symptoms can be the non-steroidal anti-inflammatory drugs (NSAIDs) and/or opioids (specifically to pain treatment). However, both groups present a large variety of side effects. For example, opioid drugs are responsible for abuse and dependence affecting 2.1 million people, and NSAIDs also increase the risk of gastrointestinal bleeding and cardiac events [2].

In Central and South America, *Aristolochia trilobata* L. (Aristolochiaceae), popularly known as “mil-homens”, is widely used in folk medicine and is an important medicinal plant [3,4]. Conditions such as stomach ache, colic, poisoning, asthma, pulmonary diseases, diabetes, and skin affections have been treated with different species from *Aristolochia* genus [5,6]. Previous studies have shown that a chloroform extract of *A. trilobata* leaves had antiphlogistic potency similar to that of indomethacin (a non-steroidal anti-inflammatory drug) [7].

The present work aims to demonstrate the traditional use of *A. trilobata* for treatment of inflammation and pain. For these purposes, we decided to obtain the essential oil of *A. trilobata* stems from enriching volatile substances, enabling a scientific study to evaluate their possible involvement in the anti-inflammatory and antinociceptive effects testing it in a well-known thermal (hot-plate) model of nociception and carrageenan-induced cell migration as a model of inflammation. We also tried to identify the mechanism by which *A. trilobata* presents its effect.

## 2. Experimental Section

### 2.1. Animals

All protocols used Swiss Webster mice (male, 20–25 g, 8–10 weeks) donated by the Instituto Vital Brazil (Niterói, Rio de Janeiro, Brazil). National Council for the Control of Animal Experimentation (CONCEA), the Biomedical Science Institute/UFRJ, and Ethical Committee for Animal Research approved the protocols used (DFBCICB015–04/16). The animals were maintained under standard conditions (a room with a 12 h light–dark cycle at 22 ± 2 °C, 60% to 80% humidity, and with food and water provided ad libitum).

### 2.2. Plant Material

Stems of *Aristolochia trilobata* were collected in Estância/SE, Brazil, in October/2011 (Geographic coordinates: S 11° 14′ 22.4′′ and W 037° 25′ 00.5′′) and received a voucher # ASE 23.161 deposited in the Herbarium of the Federal University of Sergipe. Stems of *A. Trilobata* were cut in small pieces and crushed in a fourknife mill (Marconi, model MA680). The essential oil (EO) was obtained after hydrodistillation of 200 g of stem (in 1500 mL of distilled water) along 3 h and with help of a Clevenger-type apparatus. After physically separating oil and water, the first one was dried over anhydrous sodium sulfate and filtered. EO was stored in a freezer until further analyses and assays. Identification of constituents of EO was performed as Santos and collaborators [8].

### 2.3. Drugs and Treatments

Acetylsalicylic acid (ASA), atropine sulfate monohydrate, dexamethasone, and L-nitro arginine methyl ester (L-NAME) were purchased from Sigma-Aldrich (St. Louis, MO, USA). Morphine sulfate and naloxone hydrochloride were kindly provided by Cristália (São Paulo, Brazil) and formalin was purchased from Isofar (Rio de Janeiro, Brazil). EO was dissolved in pure oil in order to prepare a stock solution (100 mg/mL). From this stock solution, intermediate solutions were prepared and administered by oral gavage at doses varying from 1 to 100 mg/kg, in a final volume of 0.1 mL of pure oil per animal. Essential oil, as well as all drugs, were diluted just before use and the pure oil used as vehicle did not present any effect per se.

### 2.4. Formalin-Induced Licking Behavior

Formalin (2.5%, μL *v*/*v*) was injected into the dorsal surface of the left hind paw of mice. The time in which animals remained licking the formalin-injected paw was recorded according to Reference [9] with some adaptations done by Matheus et al. [10]. The response was divided into two phases: the first one (neurogenic phase) occurs in the first 5 min post-formalin injection and the second one (inflammatory phase) occurs between 15 and 30 min post-formalin injection.

### 2.5. Thermal Nociception Model (Hot-Plate)

The reaction time (licked fore and hind paws) that mice remained on a hot plate (Insight Equipment, Brazil) set at 55 ± 1 °C was recorded at several intervals of 30 min post-oral administration of EO or sulcatyl acetate (1, 10, or 100 mg/kg), vehicle, morphine, or antagonists. Baseline was calculated by the mean of two reaction time measurements at 60 and 30 min before oral administration [9] adapted by Matheus et al. [10]. Area under the curve (AUC) graphs were calculated from time–course graphs. The following formula, which is based on the trapezoid rule, was used to calculate the AUC: AUC = 30 × IB [(min 30) + (min 60) +… + (min 180)]/2, where IB is the increase from the baseline (in %).

### 2.6. Evaluation of the Possible Mechanisms of Antinociception of A. Trilobata EO and Sulcatyl Acetate

The participation of nitrergic, opioid, and cholinergic pathways in the antinociception caused by EO and sulcatyl acetate was evaluated 15 min after intraperitoneal injection of antagonists and prior to oral administration of EO or sulcatyl acetate (at 100 mg/kg dose each). In assays conducted in our laboratory, dose–response curves for each antagonist against the respective agonist were previously constructed and the dose that reduced the agonist effect by 50% was chosen for the assays. Based on these data the doses used were: naloxone (opioid receptor antagonist) 1 mg/kg, atropine (cholinergic receptor antagonist) 1 mg/kg, L-NAME (nitric oxide synthase inhibitor) 3 mg/kg. The antinociceptive effect was evaluated via the hot plate test as described above.

### 2.7. Carrageenan-Induced Leukocyte Migration into the Subcutaneous Air Pouch (SAP)

Mice back received a subcutaneous injection of sterile air (10 mL) with a replacement of another 7 mL after 3 days. Twenty-four hours after the last injection of air, a solution of carrageenan (1%, 0.5 mL) was injected in subcutaneous air pouches [11,12] with modifications described in Raymundo et al. [13]. Treated groups were composed of mice that received oral administration of vehicle, EO (1, 10, or 100 mg/kg), sulcatyl acetate (1, 10, or 100 mg/kg), or dexamethasone (0.3 mg/kg, i.p.) 1 h before carrageenan injection in the SAP. The negative control group was composed of animals that received oral administration of pure oil and phosphate buffer saline (PBS, 1 mL) in SAP. After 24 h, animals were euthanized, the pouches were washed with 1 mL of sterile PBS, and exudates were collected. The total number of leukocytes was determined using a CellPocH-100Iv Diff (Sysmex) hematology analyzer. Exudates were centrifuged at 170× *g* for 10 min at 4 °C, and the supernatants were collected and stored at −20 °C until use.

To rule out a possible toxic effect of EO and sulcatyl acetate, mice treated with the highest dose of each one had their bone marrow cells collected through flushing the femur with 1 mL of PBS. Peripheral blood was also collected in heparinized tubes. The counting of the cells in the femoral lavage or in the blood was performed with the aid of a CellPocH-100Iv Diff (Sysmex) hematology analyzer.

### 2.8. Cell Culture

All cell culture reagents were purchased from Sigma-Aldrich (USA). RAW 264.7 (TIB-71) was obtained from the American Type Culture Collection. Cells were routinely grown in Roswell Park Memorial Institute (RPMI) medium containing 10% fetal bovine serum, 1% L-glutamine, and 1% penicillin-streptomycin (henceforth called RPMI) in a humidified 5% CO_2_ atmosphere at 37 °C. Cells were cultured up to confluence and used in the assays.

### 2.9. Cell Viability Assay

Cell viability was determined using 3-(4,5-dimethyl-2-thiazyl)-2,5-diphenyl-2*H*-tetrazolium bromide (MTT) reagent (Sigma-Aldrich, USA) using method described by Denizot and Lang [14]. Briefly, cells were plated at an initial density of 5 × 10^4^ cells per well in 96-well plates and incubated for 24 h at 37 °C and 5% CO_2_. After 24 h, cultures were treated with EO or SA at a final concentration of 10, 30, or 100 µg/mL and further incubated for 24 h. The supernatants were removed and then 10 μL of MTT solution (5 mg/mL in RPMI)/100 μL of medium were added to each well and incubated for 4 h at 37 °C, 5% CO_2_. The resultant formazan crystals were dissolved in dimethylsulfoxide (100 μL) and absorbance was measured in a microplate reader (FlexStation Reader, Molecular Devices, San Jose, CA, USA) at 570 nm. All experiments were performed in triplicate, and cell viability was expressed as a percentage relative to the untreated control cells.

### 2.10. Quantification of TNF-α and IL-1β

Supernatants from the exudates collected from the SAP were used to measure the levels of the cytokines tumor necrosis factor-α (TNF-α) and interleukin 1β (IL-1β) by enzyme-linked immunosorbent assay (ELISA) using the protocol supplied by the manufacturer (B&D, Franklin Lakes, NJ, USA).

### 2.11. Nitrate and Nitrite Measurement

To evaluate the nitrate accumulated in SAP, exudates were measured according to the method described by Bartholomew [15] and adapted by Raymundo et al. [13], followed by measurement of nitrite according to the Griess reaction [16].

### 2.12. Detection of Enzymes Expression

Immunoblots were carried out as described previously [17]. Briefly, RAW 264.7 cells (4 × 106/mL) were plated in 12-well plate, incubated for 1 h with EO or SA, activated with lipopolysaccharide (LPS) (1 μg/mL), and further incubated for 10 min, 1 h, or 8 h. Cells were lysed with cold lysis buffer (10% NP40, 150 Mm NaCl, 10 Mm Tris HCl pH 7.6, 2 Mm PMSF, and 5 µM leupeptin). After determination of the protein concentration in the suspensions by the BCA method (BCA™ Protein Assay Kit, Pierce, Waltham, MA, USA), the suspensions were boiled in application buffer (DTT 100 Mm, Bromophenol Blue 0.1%). Aliquots of 30 µg of protein were submitted to electrophoresis in 10% polyacrylamide gel. Proteins were electrophoretically transferred onto nitrocellulose membranes. Membranes were incubated with primary antibodies (Cell Signaling, Danvers, MA, USA) and further with secondary antibodies (anti-mouse IgG antibody conjugated to horseradish peroxidase). Proteins were detected using enhanced chemiluminescence (ECL) reagents and quantified using a ChemiDoc system (BioRad, Hercules, CA, USA).

### 2.13. Statistical Analysis

Each experimental group consisted of 6 to 8 mice, and the results are expressed as the mean ± S.D. The area under the curve (AUC) was calculated using Prism Software 5.0 (GraphPad Software, La Jolla, CA, USA). Significant differences between the groups were established using Bonferroni’s test for multiple comparisons after analysis of variance (ANOVA) testing. *p* values less than 0.05 were considered significant.

## 3. Results

### 3.1. Antinociceptive Effect

#### 3.1.1. Formalin-Induced Licking Behavior

We further decided to evaluate a possible antinociceptive activity from the EO and its major component, sulcatyl acetate. Doses of 1, 10, or 100 mg/kg given orally significantly reduced both phases of formalin-induced licking behavior (1st and 2nd). While first phase was inhibited by 10 and 100 mg/kg, the second phase was only inhibited by the highest dose (of 100 mg/kg) of EO. When studying sulcatyl acetate it could be observed that even 1 mg/kg dose significantly reduced the first phase, while only the highest dose reduced the second phase (Figure 1).

#### 3.1.2. Hot-Plate Model

The significant effect observed in the first phase of the formalin-induced licking behavior is suggestive of an antinociceptive activity. Previous data from literature had shown that EO from *A. trilobata* presented antinociceptive effect in models of chemical nociception (acetic acid-induced writhings and formalin-induced licking response). However, in such paper, authors used high doses (25, 50, and 100 mg/kg) [6]. We further decided to evaluate if lower doses of EO and its majority component (sulcatyl acetate) present antinociceptive effect in thermal model of nociception (the hot plate). We also tried to evaluate the possible mechanism of action.

Data showed in Figure 2 indicate that both EO and sulcatyl acetate presented central antinociceptive effect. Even 30 min after oral administration of EO a significant effect was observed with all three doses tested (1, 10, and 100 mg/kg). This effect was maintained until 90 min post-oral administration and being gradually reduced in later times. The conversion of data from line graphs to area under the curve graph demonstrated that all three doses tested developed a significant central antinociceptive effect when comparing to the vehicle-treated group.

When mice were orally treated with sulcatyl acetate, we could observe a significant effect after 30 min of administration. Differently to that observed with EO, sulcatyl acetate antinociception was maintained until 150 min. At this time point, antinociceptive effect was even higher than morphine-treated mice. The graph of the area under the curve demonstrated that all 3 doses presented a significant antinociceptive effect.

#### 3.1.3. Mechanism of Antinociceptive Action

In the next step, we decided to investigate which pathways could be involved in the central antinociceptive effect of EO and sulcatyl acetate. Mice were pretreated with an opioid antagonist (naloxone), a cholinergic antagonist (atropine), or with an inhibitor of nitric oxide synthase enzyme (L-NAME). Data obtained showed that none of the antagonists of inhibitor significantly reversed the antinociceptive effect of either EO or sulcatyl acetate (Figure 3).

### 3.2. Anti-Inflammatory Effect

The observation that EO and sulcatyl acetate inhibited the second phase of formalin-induced licking, a well-known inflammatory phase, led us to investigate if both substances could present an anti-inflammatory effect. In this regard, EO and sulcatyl acetate were evaluated in their abilities in reducing carrageenan-induced leukocyte migration into a subcutaneous air pouch (SAP) and cytokines production.

#### 3.2.1. Leukocyte Migration

Injection of carrageenan into the SAP led to a 76-fold increase in leukocyte number (2.14 ± 1.65 × 10^6^ cells/mL in the group that received saline in SAP versus 162.6 ± 31.17 × 10^6^ cells/mL in the group that received carrageenan in SAP). Pretreatment of mice with the steroidal anti-inflammatory drug (SAID), dexamethasone (Dex) resulted in a reduction of 50% in leukocyte number present in SAP. The crescent doses of EO (1, 10, and 100 mg/kg) significantly reduced the cell migration with values similar to the SAID. Sulcatyl acetate also reduced the number of cells that migrated to the pouch. However, this effect was less prominent when compared with that obtained with EO (Figure 4).

#### 3.2.2. Protein Extravasation

As EO and SA presented a significative effect in reducing leukocyte migration into SAP similarly to that observed with the positive control group (SAID), we decided to further investigate if they could reduce other parameters of the inflammatory process in a tentative to investigate the possible mechanism of anti-inflammatory action. The exudate collected in the SAP protein was measured and results demonstrated that pretreatment of mice with 10 or 100 mg/kg doses of EO significantly reduced the amount of protein in exudate. It is interesting to note that none of the doses of SA inhibit protein extravasation even at a higher dose (100 mg/kg) (Figure 5).

#### 3.2.3. Nitric Oxide Production

We also decided to quantify the amount of nitric oxide (NO) produced in the exudate. When NO is produced in biological fluids it decays to the stable metabolite nitrate. Figure 6 shows that injection of carrageenan in the SAP lead to an increase in the amount of NO accumulated in the pouch when compared with mice that received saline. EO almost completely abolished NO production resulting in NO levels similar to those observed in saline-treated mice. Although SA did not completely inhibit the NO production, the reduction observed vary between 50% and 80% (Figure 6).

#### 3.2.4. Cytokine Production

The measurement of cytokines TNF-α and IL-1β accumulated in the exudate obtained from SAP showed that highest doses of EO (10 and 100 mg/kg) significantly reduced levels of both cytokines. However, pretreatment of mice with the majority component of the EO, sulcatyl acetate, led to an almost 50% reduction in cytokines production even with 1 mg/kg dose (Figure 7).

#### 3.2.5. In Vitro Cell Viability and Nitric Oxide Production

To further evaluate the possible anti-inflammatory mechanism of action, we decided to study the effects of EO and SA in vitro using macrophage cell line RAW 264.7. Data obtained showed that none of the concentrations used (1, 10, or 30 µg/mL) significantly affected the cell viability (data not shown). We next measure NO production by LPS-activated cells incubated or not with the three concentrations of EO and SA. Neither EO nor SA did induce NO production *per se* (data not shown). Figure 8 shows that there is an inhibitory effect on NO production when LPS-activated cells were pre-incubated with SA for 1 h (Figure 8D). In another set of experiments, cells were activated with LPS and after 8 h different concentrations of OE or SA were added to each group. As can be observed in Figure 8C,D, we do not observe inhibitory effect in NO production when EO or SA was added 8 h post-LPS.

In a tentative to demonstrate the possible mechanism of anti-inflammatory effect of EO and SA, we decided to evaluate their effects on some inflammatory pathways. For this purpose, expression of inducible nitric oxide synthase (iNOS), p38 mitogen-activated protein kinase (p38 MAPK), its activated form (the phosphorylated p38), and spleen tyrosine kinase (Syk) was evaluated after incubation of RAW 264.7 cells with EO or SA (30 µg/mL) and LPS. As expected, non-activated cells did not express those inflammatory enzymes, while LPS incubation induced an increase in levels of all of them. As illustrated in Figure 9, preincubation with EO significantly reduced the expression of iNOS. To ascertain whether the mechanism by which both substances were reducing the expression of the enzyme, we evaluated the involvement of enzymes present in the early stages of the activation pathways triggered by LPS in the toll receptor 4. In this regard, we evaluated expression of p38 MAPK and Syk and the corresponding increase in levels of phosphorylated p38, a consequence of cell activation and p38 activation. It can be observed that EO reduced expression of Syk enzyme. After an activation induced by LPS we can observe the activation/phosphorylation of p38 MAPK. In this context, it could be noted an increase in levels of this enzyme (p-p38). However, preincubation of activated cells with EO or SA did not affect phosphorylation levels of p38 MAPK. In summary, EO and SA did not affect the levels of p-p38 expressed in cells after activation with LPS (Figure 9).

## 4. Discussion

The present work demonstrated that the essential oil (EO) of *A. trilobata* and its majoritarian substance, sulcatyl acetate (SA), present significant anti-inflammatory and antinociceptive effects. We also demonstrated that opioidergic, nitrergic, and cholinergic pathways do not participate in the antinociceptive effect. Contributing to the knowledge about this species, we also demonstrated that both EO and SA presented anti-inflammatory effect reducing leukocyte migration, production of cytokines, nitric oxide (NO), and protein extravasation.

The evaluation of *A. trilobata* EO chemical constituents highlighted the presence of SA as the major substance. This characteristic is not usual in essential oils in general. However, this finding confirms previous works, which states *A. trilobata* EO as a rich source of SA [4,7].

Quintas et al. [7] demonstrated that EO from *A. trilobata* and SA presented antinociceptive effect in two models of chemical nociception (i.e., acetic acid-induced writhing and formalin-induced paw licking). Although the essential oil tested in that work was very similar to that used by us, there are several differences among assays and results. In that work, authors used doses varying from 25 to 100 mg/kg, administered by intraperitoneal route. EO and SA were evaluated only in two models of chemical nociception. Differently, in the present work, we evaluated EO and SA administered by oral route and doses of 1, 10, and 100 mg/kg. The effect observed in this model could be explained, at least in part, due to the presence of other substances with antinociceptive activity. It was demonstrated by Amaral and collaborators [18] that the monoterpene limonene presented antinociceptive activity more related to peripheral analgesia. Kaiamoto and collaborators [19] also suggested that limonene may act as a transient receptor potential cation channel 1 (TRPA1) agonist when applied topically. However, when systemically administered, it presents an antinociceptive effect. In our work, we did not test isolated limonene. We used only the majoritarian substance (SA). It is also important to note that formalin-induced licking is a multicomplex phenomenon with the involvement and activation of nuclear factor erythroid 2-related factor 2 (Nrf2) pathway. Nrf2 is a transcriptional factor related to activation of heme-oxygenase 1 (HO-1) [20], leading to an antinociceptive effect in formalin-induced licking [21]. It may be that in such a way, by oral administration, all the substances presented in the EO may acting together to present an anti-inflammatory effect.

Considering the *A. trilobata* EO results and its major substances, a synergistic effect of SA and limonene can be occurring, especially in the second phase of formalin-induced licking response. Besides evaluating the effects in the formalin-induced paw licking, we also used the thermal model of nociception, the hot-plate, and searched the mechanism of action of EO and SA using three different antagonists.

In another group of assays, we also studied the anti-inflammatory effect of EO and SA in their capacity to inhibit the leukocyte migration into the subcutaneous air pouch induced by carrageenan and production of some inflammatory mediators. Our data complement those obtained by the other group and identify the possible mechanism of action. We also demonstrated that EO and SA presented significant effects even when used by oral route at a dose as low as 1 mg/kg.

In the present study, we attempted to further characterize some of the mechanisms through which EO from *A. trilobata* and SA exerts its antinociceptive effect. The sensation of pain can be divided into four components, i.e., transduction, transmission, modulation, and perception. Several systems, such as oxidonitrergic, gamma aminobutyric acid GABAergic, glutamatergic, opioidergic, cholinergic, serotonergic, adrenergic, and others, may act in different steps of the pain turning it a complex phenomenon in such a way that alterations in one of the system cited above can alter the response in all the other systems [22]. A substance, whether natural or synthetic, can then act in one of four components to produce analgesia. Our data are suggestive that nor opioidergic, nitrergic, or cholinergic pathways appear to be involved in the antinociceptive effect of EO and SA since none of the antagonists used (naloxone, L-NAME, or atropine) reversed the effects of EO and SA. Despite the opioid system is one of the most important in pain perception and modulation, the activation of naloxone-sensitive pathway is probably not involved in the antinociception produced by EO and SA because naloxone significantly reversed morphine (data not shown) but not EO and SA antinociception.

We also demonstrated that doses as low as 1 mg/kg significantly reduced the licking response induced by formalin injection in mice paw. Nociception induced by formalin develops a biphasic pattern with an initial phase (5 to 15 min post-injection) followed by a second phase (15 to 30 min post-injection) [23,24,25,26]. First phase is due to direct activation of nociceptors, whereas the second one is due to the release of inflammatory mediators acting together in nociceptors and their own local receptors [27,28,29]. The involvement of serotonin and bradykinin in both phases was also described [30].

As mediators previously cited are also involved in inflammatory processes, we further decided to study if EO and SA could present an anti-inflammatory effect. In this regard, we used the leukocyte migration induced by injection of carrageenan into the subcutaneous air pouch (SAP). Carrageenan-induced inflammation is a multicomplex phenomenon with the involvement of synthesis and/or liberation of a range of mediators such as prostaglandins, histamine, bradykinin, serotonin, nitric oxide, and leukotrienes and chemotaxis of neutrophils and macrophages [31]. After 24 h of carrageenan injection, there is an intense migration of leukocytes and an increase in levels of NO and cytokines [32,33,34,35,36].

Results obtained in the SAP model complement those obtained in the second phase of formalin-induced licking since both models present an inflammatory profile with involvement in a diversity of inflammatory mediators. One hypothesis for the reduction observed in this model could be related to reduction in production and/or liberation of inflammatory substances involved in leukocyte chemotaxis. In this context, we can infer that the reduction in cell migration into the SAP could be due to a reduction in the levels of these cytokines. Our data are in accordance with those obtained in Reference [37], which observed that tacrolimus (an immunosuppressor) reduced neutrophil infiltration in the pancreatitis model due to a reduction in expression of mRNA of TNF-α and IL-1β. In an inflammatory event, inducible nitric oxide synthase is expressed in different cells and culminating with NO production. NO plays multiple roles in the inflammatory response, vasodilation, and regulation of leukocytes rolling, migration, cytokines production, and proliferation. It has been shown that some iNOS inhibitors demonstrated an important effect in several inflammatory models, such as the air pouch model [38,39]. The fact that EO and SA reduced NO levels in exudate also contribute to their anti-inflammatory effect. It is also interesting to note that EO reduced NO levels similar to the positive control group, dexamethasone, a well-known steroidal anti-inflammatory drug.

Data obtained using the in vivo model of SAP was corroborated by in vitro assays. Reduction in nitric oxide production in vitro and in vivo was not due to a direct effect in iNOS enzyme activity. It is well known that iNOS synthesis occurs until 6–8 h after LPS activation. After this time point, nucleus synthesis is finished and enzymes synthesized initiate NO production [39]. Since addition of EO or SA 8 h after LPS activation did not affect NO production, it can be suggested that the inhibitory effect observed was not due to reduced enzyme activity.

As a final step in our effort to delineate the inhibitory effects of EO and SA, we quantified Syk and p38 MAPK enzymes expression. Several intracellular signaling molecules are involved and activated during the inflammatory responses in macrophages. Tyrosine kinase families have been considered as the major effector molecule. Spleen tyrosine kinase (Syk) binds with Toll-like receptor-4 (TLR4) and is activated, resulting in the transduction of stimulatory signals through the activation of various downstream signaling molecules. Since Syk is one of the upstream signaling molecules, it orchestrates many downstream signaling molecules and amplifies inflammatory signals. Therefore, Syk has been considered to play critical roles in inflammatory responses [40,41,42]. The observation that EO partially reduced expression of Syk can explain the inhibition of expression of iNOS and justify the inhibition observed in NO production.

p38 MAPK was first recognized for its role in inflammation in regulating the biosynthesis of pro-inflammatory cytokines (i.e., IL-1 and TNFα) in LPS-stimulated cells [41] and expression of cycloxigenase-2 (COX2) [42,43,44]. It can be suggested that EO and SA mechanism of action do not seem to affect p38 MAPK activation into the phosphorylated form (the p-p38). We could speculate that EO effect may act through Syk pathway do not interfere with p38 MAPK since both are independent systems that may act without cross-interference with each other.

Our data also demonstrated that SA did not affect *per se* any of enzymes evaluated (Syk, p38 MAPK, and iNOS). It is important to note that as part of a complex mixture of substances it is reasonable that EO can present effect and one isolated substance cannot present the same effect and mechanism of action.

## 5. Conclusions

In the present work, we demonstrated that *A. trilobata* essential oil as well as the majority component, sulcatyl acetate, present antinociceptive and anti-inflammatory effects. This activity is accompanied by reduction in cell migration and production of NO and cytokines. It seems that at least part of this effect is mediated by inhibition of Syk and iNOS enzymes expression. Together, we can suggest *A. trilobata* as an anti-inflammatory and antinociceptive species.

## Figures and Tables

**Figure 1 biomedicines-08-00111-f001:**
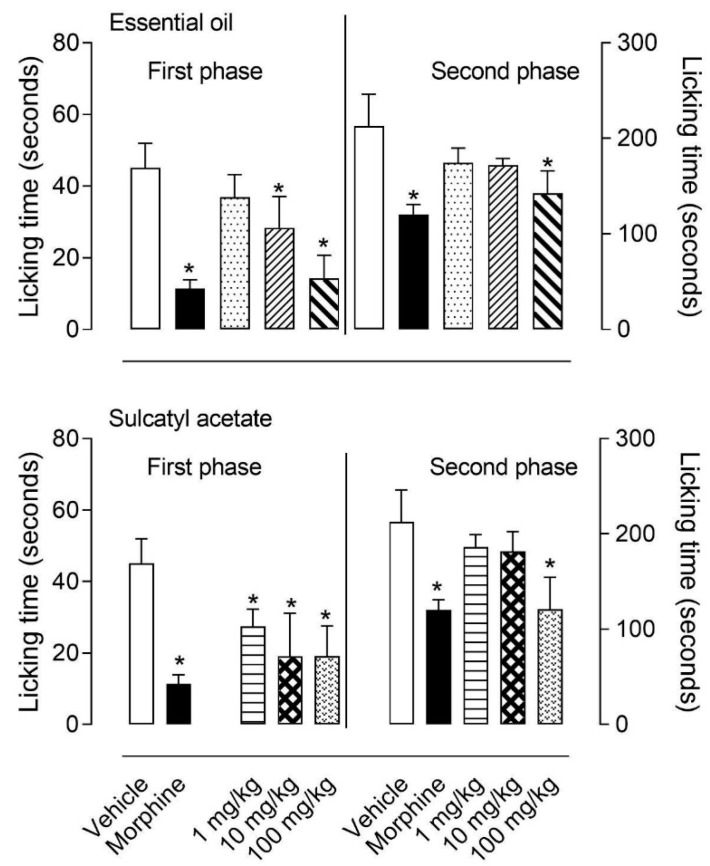
Effect of essential oil from *Aristolochia trilobata* and sulcatyl acetate on formalin-induced licking response in mice. Animals were orally pretreated with essential oil of sulcatyl acetate (1, 10, or 100 mg/kg), acetylsalicylic acid (ASA, 200 mg/kg), morphine (2.5 mg/kg), or vehicle (DMSO + phosphate buffer saline (PBS) 60 min before intraplantar injection of formalin (2.5%). The results are expressed as the mean ± S.D. (*n* = 6) of the time the animals spent licking the formalin-injected paw. Statistical significance was calculated by one-way ANOVA with Bonferroni’s as post-test. * indicates *p* < 0.05 when compared to the vehicle-treated mice.

**Figure 2 biomedicines-08-00111-f002:**
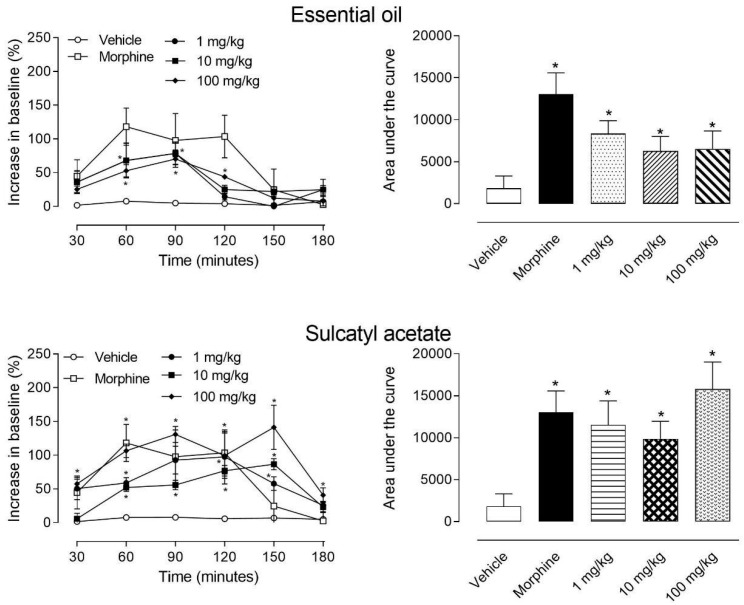
Effect of essential oil from *A. trilobata* and sulcatyl acetate on thermal nociception model (the hot plate) in mice. Animals were orally pretreated with essential oil of sulcatyl acetate (1, 10, or 100 mg/kg), morphine (2.5 mg/kg), or vehicle (DMSO + PBS). The results are presented as the mean ± S.D. (*n* = 6–8) of the increase in the response time relative to baseline levels (left graphs) or area under the curve (right graphs) calculated with the Prism Software 5.0. Statistical significance was calculated by one-way ANOVA with Bonferroni’s as post-test. * indicates *p* < 0.05 when compared to the vehicle-treated mice. Where no error bars are shown is because they are smaller than the symbol.

**Figure 3 biomedicines-08-00111-f003:**
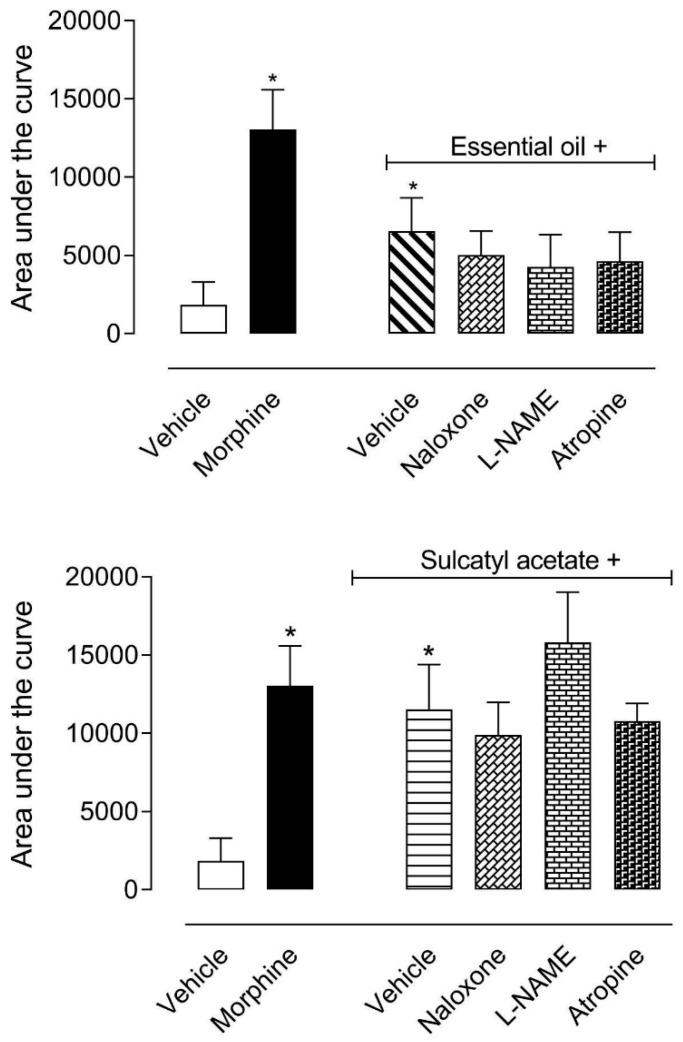
Effect of the different antagonists on the antinociceptive activity of essential oil from *A. trilobata* and sulcatyl acetate in the hot-plate model. The animals were pretreated with naloxone (1 mg/kg, i.p.), atropine (1 mg/kg, i.p.), or L-NAME (3 mg/kg, i.p.) 15 min before oral administration of essential oil or sulcatyl acetate (100 mg/kg) or vehicle. The results are expressed as the mean ± S.D. of the area under the curve calculated with Prism Software 5.0 (*n* = 6–8). One-way ANOVA followed by Bonferroni’s test was used to calculate the statistical significance. * indicates *p* < 0.05 when compared to the vehicle-treated group.

**Figure 4 biomedicines-08-00111-f004:**
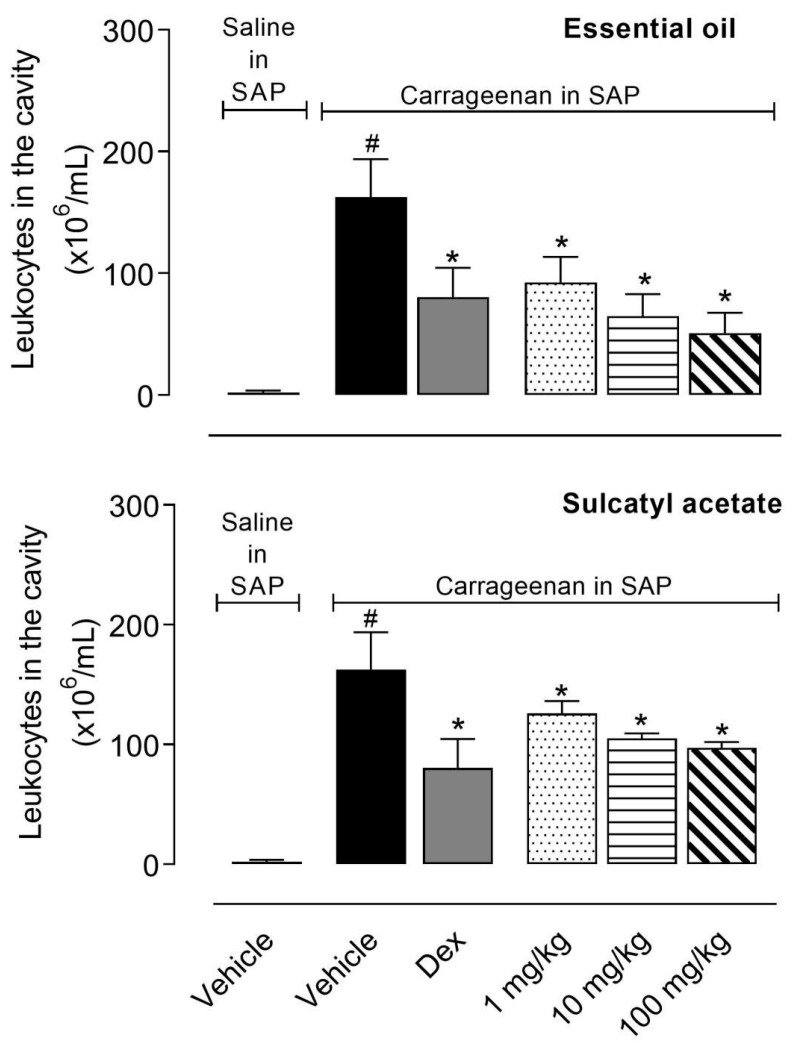
Effect of essential oil from *A. trilobata* and sulcatyl acetate in leukocyte migration into the subcutaneous air pouch (SAP). The animals were pretreated with dexamethasone (Dex, 1 mg/kg, i.p.), essential oil, or sulcatyl acetate (1, 10, or 100 mg/kg) or vehicle, 1 h before carrageenan injection into the SAP. The results are expressed as the mean ± S.D. calculated with Prism Software 5.0 (*n* = 6–8). One-way ANOVA followed by Bonferroni’s test was used to calculate the statistical significance. * indicates *p* < 0.05 when compared to the vehicle-treated group (animals that received carrageenan in the SAP) and # indicates *p* < 0.05 when compared to the vehicle-treated group (animals that received saline in the SAP).

**Figure 5 biomedicines-08-00111-f005:**
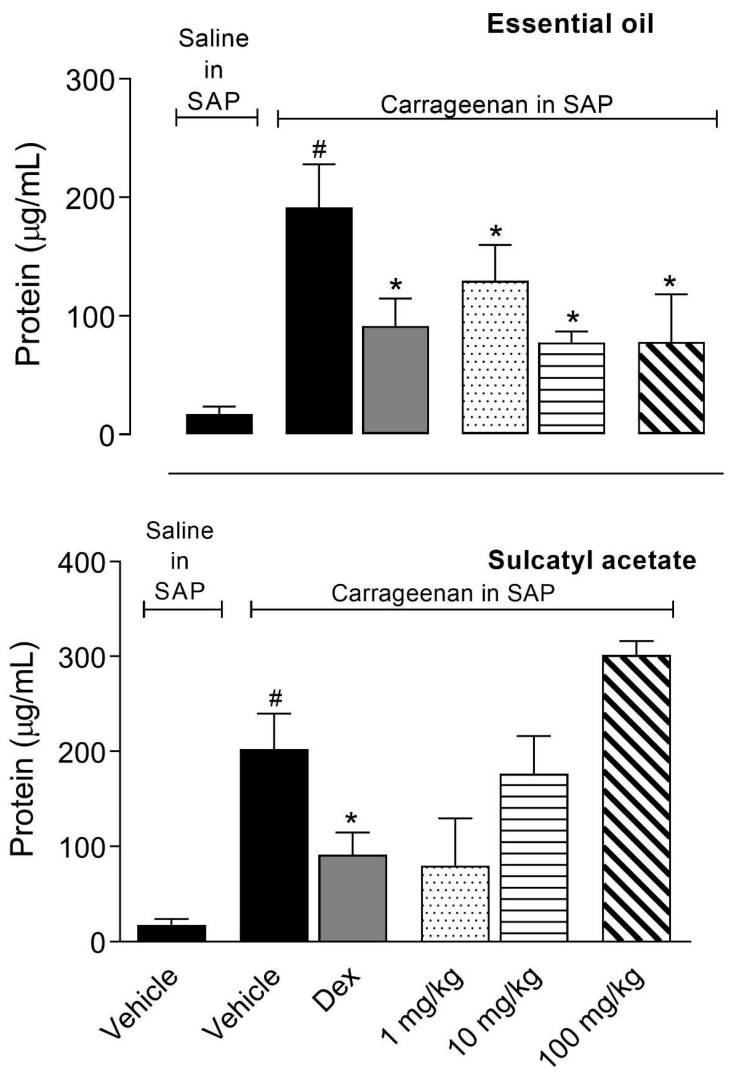
Effect of essential oil from *A. trilobata* and sulcatyl acetate in protein extravasation into the subcutaneous air pouch (SAP). The animals were pretreated with dexamethasone (Dex, 1 mg/kg, i.p.), essential oil, or sulcatyl acetate (1, 10, or 100 mg/kg) or vehicle 1 h before carrageenan injection into the SAP. The results are expressed as the mean ± S.D. calculated with Prism Software 5.0 (*n* = 6–8). One-way ANOVA followed by Bonferroni’s test was used to calculate the statistical significance. * indicates *p* < 0.05 when compared to the vehicle-treated group (animals that received carrageenan in the SAP) and # indicates *p* < 0.05 when compared to the vehicle-treated group (animals that received saline in the SAP).

**Figure 6 biomedicines-08-00111-f006:**
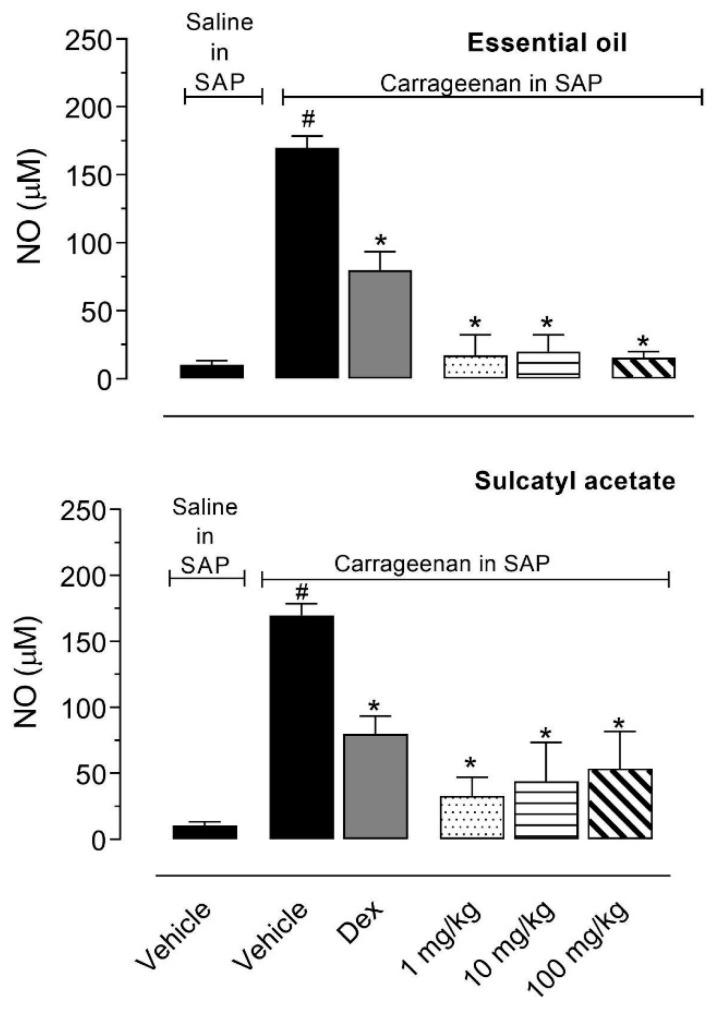
Effect of essential oil from *A. trilobata* and sulcatyl acetate in nitric oxide production into the subcutaneous air pouch (SAP). The animals were pretreated with dexamethasone (Dex, 1 mg/kg, i.p.), essential oil, or sulcatyl acetate (1, 10, or 100 mg/kg) or vehicle 1 h before carrageenan injection into the SAP. The results are expressed as the mean ± S.D. calculated with Prism Software 5.0 (*n* = 6–8). One-way ANOVA followed by Bonferroni’s test was used to calculate the statistical significance. * indicates *p* < 0.05 when compared to the vehicle-treated group (animals that received carrageenan in the SAP) and # indicates *p* < 0.05 when compared to the vehicle-treated group (animals that received saline in the SAP).

**Figure 7 biomedicines-08-00111-f007:**
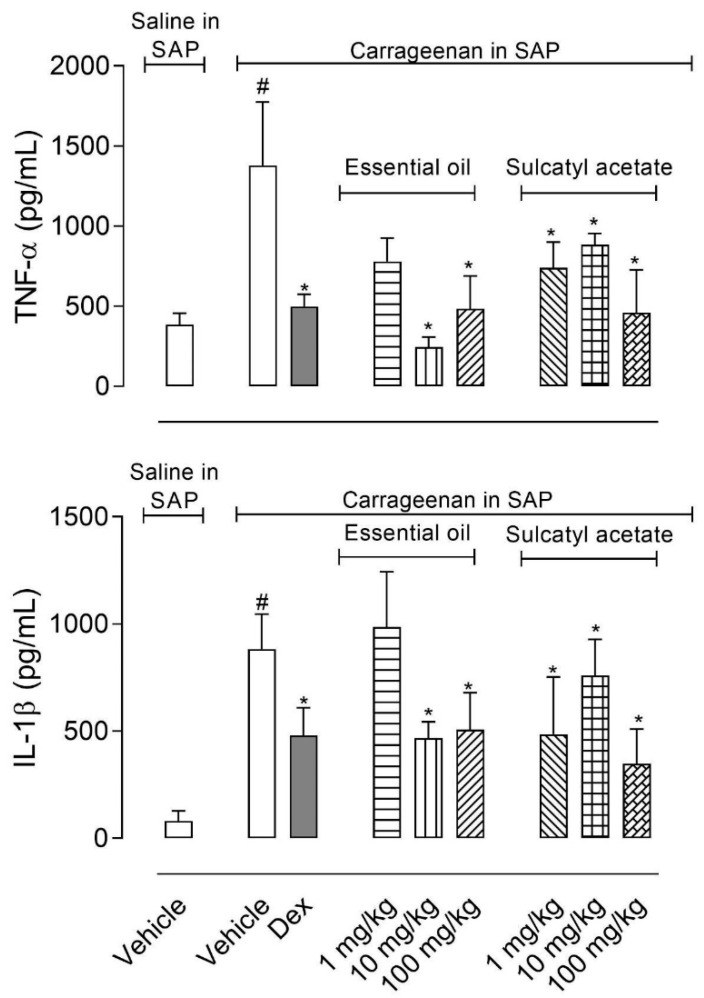
Effect of essential oil from *A. trilobata* and sulcatyl acetate in TNF-α and IL-1β production in the subcutaneous air pouch (SAP). The animals were pretreated with dexamethasone (1 mg/kg, i.p.), essential oil, or sulcatyl acetate (1, 10 or 100 mg/kg) or vehicle 1 h before carrageenan injection into the SAP. The results are expressed as the mean ± S.D. calculated with Prism Software 5.0 (*n* = 6–8). One-way ANOVA followed by Bonferroni’s test was used to calculate the statistical significance. * indicates *p* < 0.05 when compared to the vehicle-treated group (animals that received carrageenan in the SAP) and # indicates *p* < 0.05 when compared to the vehicle-treated group (animals that received saline in the SAP).

**Figure 8 biomedicines-08-00111-f008:**
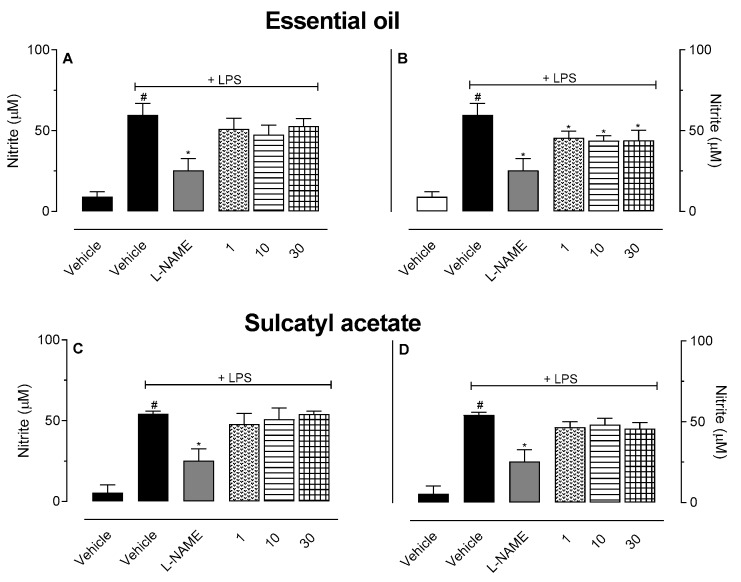
Effect of essential oil from *A. trilobata* (EO) and sulcatyl acetate (SA) in nitric oxide production by RAW 264.7 macrophage. Cells (2 × 10^6^/mL) were incubated with lipopolysaccharide (LPS, 1 µg/mL) and EO or SA (1, 10, or 30 µg/mL). Graphs **A** and **B** show the results of pre-incubation of cells for 1 h before LPS addition to cells. Graphs **C** and **D** show the results of NO production when EO or SA was added 8 h post-LPS activation. The results are expressed as the mean ± S.D. calculated with Prism Software 5.0 (*n* = 6). One-way ANOVA followed by Bonferroni’s test was used to calculate the statistical significance. # indicates *p* < 0.05 when compared to the vehicle (non-LPS group). * indicates *p* < 0.05 when compared to the group that was activated with LPS and received vehicle.

**Figure 9 biomedicines-08-00111-f009:**
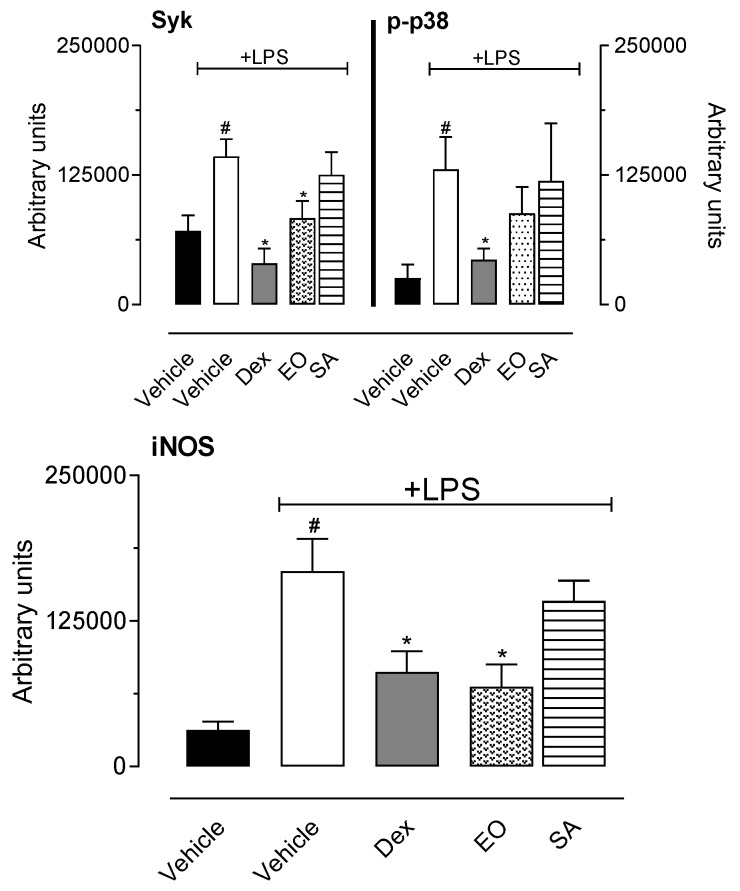
Effect of essential oil from *A. trilobata* (EO) and sulcatyl acetate (SA) in expression of Syk, p38 MAPK, or inducible nitric oxide synthase (iNOS) by RAW 264.7 cells. Cells (2 × 10^6/mL^) were incubated with dexamethasone (350 µg/mL), EO, or SA (30 µg/mL) 1 h before lipopolysaccharide (LPS, 1 µg/mL). After 10 min (for Syk and p38 MAPK) or 8 h (for iNOS), cell lysates were collected for Western blot analyses. The results are expressed as the mean ± S.D. calculated with Prism Software 5.0 (*n* = 3). One-way ANOVA followed by Newman’s test was used to calculate the statistical significance. # indicates *p* < 0.05 when compared to the vehicle (non-LPS group). * indicates *p* < 0.05 when compared to the group that was activated with LPS and received vehicle.

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
