# Peer review of "Aristolochia trilobata: Identification of the Anti-Inflammatory and Antinociceptive Effects"

_biomedicines, 2020, doi:10.3390/biomedicines8050111_

Round 1

Reviewer 1 Report

Da Costa Salome et al. have chemically analysed and studied Essential oil (EO) and sulcatyl acetate (SA) from Aristolochia trilobata in animal pain models. The authors were able to replicate a previous study in which EO and SA showed efficacy in formalin model. However, assumption in the current study is that all effects elicited by EO and SA are solely mediated by studied compounds and no data is provided or commented whether any of the observed effects could be mediated by pharmacologically active metabolites. Further, the authors have tried to elucidate underlying mechanism of action for the observed anti-inflammatory and analgesic effects by using naloxone, atropine and L-NAME but without success.

Chemical structure of SA does not resemble known opioid, muscarinic or nitrergic ligands, but have a double bond suggesting chemical reactivity. A previous study showed that limonene another reactive compound isolated from EO, act as a desensitising TRPA1 agonist (Kaimoto et al. Eur J Pain Aug;20(7):1155-65). Further, SA was shown to increase dose-dependently protein extravasation, suggesting peptidergic sensory neuron activation and neuropeptide release. Moreover, TRPA1 ion channel activation by formalin is an established mechanism of action for formalin-induced sustained pain and therefore TRPA1 antagonism is a logical candidate for any compound showing analgesic action in the formalin model.

The authors are requested to study whether SA is also a TRPA1 agonist and is able elicit nocifensive behaviour upon injection to the paw and to rewrite manuscript from the point of view that EO and SA may modulate TRPA1 ion channel function similar to limonene. Another possible mechanism of action that is relevant for pain modulation by reactive compounds is Nrf2 activation. Please comment this also in the discussion and/or provide new data.

Minor points:

Line 48: plat => plant

Line 162: explain 5 X 104 cells per well

Fig. 2: Sulcatil acetate => sulcatyl acetate

Line 395: lower => low

Line 409: lower=> low

Line 439: in vivo and in vivo => in vitro and in vivo

Author Response

  • A previous study showed that limonene another reactive compound isolated from EO, act as a desensitising TRPA1 agonist (Kaimoto et al. Eur J Pain Aug;20(7):1155-65).

In this paper, the conclusion was “...Topically applied limonene stimulates TRPA1, resulting in elicitation of acute pain, but its systemic application inhibits nociception induced by oxidative stress...”.

In our paper we did an oral administration of an essential oil. Only 15% of limonene was identified in it. It may be that in such a way, by oral administration all the substances presented in the EO may acting together to present an anti-inflammatory effect.

We added a text in discussion section. Please see lines 497-501.

  • Further, SA was shown to increase dose-dependently protein extravasation, suggesting peptidergic sensory neuron activation and neuropeptide release.

In our data, figure 6, we did not observe increase in protein extravasation. Neither essential oil nor sulcatyl acetate increased. We must compare results between mice pretreated with substances and mice pretreated with vehicle.

  • Moreover, TRPA1 ion channel activation by formalin is an established mechanism of action for formalin-induced sustained pain and therefore TRPA1 antagonism is a logical candidate for any compound showing analgesic action in the formalin model.

We also know that in formalin-induced licking we observe liberation of serotonin and histamine, mainly in the first phase (Parada et al., 2001). So, in this model we have a multitude of mediators and not only one system. It was the reason to study both substances in several models.

  • The authors are requested to study whether SA is also a TRPA1 agonist and is able elicit nocifensive behaviour upon injection to the paw and to rewrite manuscript from the point of view that EO and SA may modulate TRPA1 ion channel function similar to limonene.

This suggestion is interesting. But there is a methodological/technical problem. As na essential oil, its vehicle is tween. In this regard, if we inject tween in mouse paw we do observe an intense edema and increase in linking behaviour due to vehicle injection.

We did not test isolated limonene in any of the four models. This is the reason for do not indicate any result. Another possibility is that, as we are testing an essential oil, it is possible that not only limonene is responsible by the results.

We can observe a complex mixture of results due to a presence of a diversity of substances each one responsible for a small part of the effects observed.

  • Another possible mechanism of action that is relevant for pain modulation by reactive compounds is Nrf2 activation. Please comment this also in the discussion and/or provide new data.

We did not perform any assay measuring Nrf2 pathway. But we added a phrase in discussion section. Please see lines 449-457

Minor points:

  • Line 48: plat => plant

The word was corrected

  • Line 162: explain 5 X 104 cells per well

The correct is 5 x 104 cells per well. Probably is was a typographical error.

Now it is in line 181

  • 2: Sulcatil acetate => sulcatyl acetate

corrected

  • Line 395: lower => low

We changed the word. Now it is in line 516

  • Line 409: lower=> low

We changed the word. Now it is in line 530

  • Line 439: in vivo and in vivo => in vitro and in vivo

We changed the word. Now it is in line 561

Reviewer 2 Report

I read the manuscript with interest. The research topic is undoubtedly important and relevant however some moments are not clear to me. I have the following comments:

Abstract

Line 20. It should be pointed full name of plant object (Aristolochia trilobata L.)

Line 23. 6-methyl-5-hepten-2-yl acetate is the correct name of compound (not «6-methyl-5-hepten-2-yl»).

Line 23. «EO and SA (10, 30 and 100 mg/kg, p.o.) were evaluated…..» Strange concentrations. Other concentrations were used in these models (1, 10, 100 mg/kg).

Line 27-28. «Major components of EO were identificated as limonene (15.43%) and sulcatyl acetate (23.31%)». This sentence should be deleted. See the explanation below.

Introduction 

Line 40. «This phenomenon presents the four cardinal signs (i.e., redness, heat, swelling and pain)». It is known that five classical signs of inflammation are heat, pain, redness, swelling, and loss of function. Not four.

Line 48. «….medicinal plat». Please correct.

Line 50. Please use italics (Aristolochia genus).

Line 53. «In our continuous search for new anti-inflammatory and analgesic substances…» Please provide links.

Line 55. «… we decided to obtain the essential oil of A. trilobata stems from enriching volatile substances enabling a scientific study ».

Introduction data state that the chloroform fraction from the leaves of A. trilobata has an anti-inflammatory effect. However, you have chosen to study the essential oil from the stems. Why? What is the connection?

It is necessary to provide evidence of the chemical identity of chloroform fractions from leaves and stems. Then it is necessary to prove that the components of the essential oil are part of the chloroform fraction. The chloroform fraction is a multicomponent mixture in which compounds of various structures are present. For example, the usual components of the chloroform fraction are triterpene aglycones, which have high anti-inflammatory activity.

Why do you get the essential oil by hydrodistillation rather than chloroform extraction?

Experimental Section

Line 69. «Samples of Aristolochia trilobata…» Stems. Not samples.

Line 69. Please indicate the geographic coordinates of the place of collection of A. trilobata stems.

Results

Line 197. Section 3.1. «Chemical analysis» repeats the information presented in your previous article «Volatile constituents of Aristolochia trilobata L. (Aristolochiaceae): A rich source of sulcatyl acetate» (http://dx.doi.org/10.5935/0100-4042.20140163). Why do you duplicate the data? Table 1 is identical to the table in the above article.

I also recommend you delete chemical information from the Experimental section 2.2 and 2.3 since these data were previously presented. 

Line 211. The phase designations should correspond to each other in the Experimental section and in the Results Section («first phase, second phase… 1st phase, 2nd phase».

Line 228. Please check the link. I think the information from link 7 is presented.

Line 255. I do not see the data on the essential oil in the Figure 4. There is information only about sulcatyl acetate.

Figure 4. Why L-NAME increases the effectiveness of sulcatyl acetate as an antinociceptive agent, although a decrease in activity would be expected, which is observed for other inhibitors (naloxone, atropine)?

Figure 5. I am confused by the description of the results in section 3.3.1. In my opinion according Figure 5, the best reduction in leucocyte number was observed for essential oil. Describe your results.

Figure 5. Where is data on dexamethasone? How can you reason about the results of an experiment without data on the action of the reference substance?

Line 284. «As EO and SA presented significative effect reducing leukocyte migration…» Compared to what, the effect was significative?

Figure 6, section 3.3.2. There is no reference substance. I cannot evaluate the data obtained.

Figure 7, section 3.3.3. There is no reference substance. I cannot evaluate the data obtained.

Figure 8, section 3.3.4. There is no reference substance. I cannot evaluate the data obtained.

Figure 9, section 3.3.5. It makes no sense to discuss the data. There is no correct description of the location of groups of essential oil and sulcatyl acetate. No data for reference substance.

Figure 10. What does it mean – «p-p38»? There is no reference substance. I cannot evaluate the data obtained. Why did you use a single dose in this experiment (30 µg/mL)? Before that you used 3 different concentrations.

Discussion

Line 369. «The present work demonstrated that the essential oil of A. trilobata and its majoritarian substance, sulcatyl acetate, present significant anti-inflammatory and antinociceptive effects». I can’t talk about the significance of the anti-inflammatory and antinociceptive effect without data on the effect of the reference compounds.

Line 395. «We also demonstrated that EO and SA presented significant effect even when used by oral route a dose as lower as 1 mg/kg». I did not find in the manuscript data on the initial concentration less than 1 mg/kg.

Line 409. «We also demonstrated that doses as lower as 1 mg/kg significantly reduced the licking response…». I did not find in the manuscript data on the initial concentration less than 1 mg/kg.

References

Please check links 7 and 10.

I cannot recommend this manuscript for publication. I don’t understand the selection of the object of study, the duplication of data. Moreover, I cannot agree or refute the Discussion part of the authors, since the results of experiments do not provide comparative information on reference compounds, which is a gross violation.

Author Response

Abstract

-Line 20. It should be pointed full name of plant object (Aristolochia trilobata L.)

We added the full name

-Line 23. 6-methyl-5-hepten-2-yl acetate is the correct name of compound (not «6-methyl-5-hepten-2-yl»).

We corrected the name

-Line 23. «EO and SA (10, 30 and 100 mg/kg, p.o.) were evaluated…..» Strange concentrations. Other concentrations were used in these models (1, 10, 100 mg/kg).

Reviewer is correct. It was our mistake. The correct doses are 1, 10 and 100 mg/kg. We decided to use a log scale for concentrations.

-Line 27-28. «Major components of EO were identificated as limonene (15.43%) and sulcatyl acetate (23.31%)». This sentence should be deleted. See the explanation below.

As indicated by Reviewer we removed this sentence and all data concerning chemical identification. Sections 2.2 and 2.3

 Introduction 

-Line 40. «This phenomenon presents the four cardinal signs (i.e., redness, heat, swelling and pain)». It is known that five classical signs of inflammation are heat, pain, redness, swelling, and loss of function. Not four.

We corrected the text and added the 5th signal. Please see line 40-41.

-Line 48. «….medicinal plat». Please correct.

We corrected the text. Please see line 47.

-Line 50. Please use italics (Aristolochia genus).

We corrected the text. Please see line 49

-Line 53. «In our continuous search for new anti-inflammatory and analgesic substances…» Please provide links.

We removed this sentence. Please see line 52.

-Line 55. «… we decided to obtain the essential oil of A. trilobata stems from enriching volatile substances enabling a scientific study ».

Introduction data state that the chloroform fraction from the leaves of A. trilobata has an anti-inflammatory effect. However, you have chosen to study the essential oil from the stems. Why? What is the connection? It is necessary to provide evidence of the chemical identity of chloroform fractions from leaves and stems. Then it is necessary to prove that the components of the essential oil are part of the chloroform fraction. The chloroform fraction is a multicomponent mixture in which compounds of various structures are present. For example, the usual components of the chloroform fraction are triterpene aglycones, which have high anti-inflammatory activity.

The citation of the anti-inflammatory effect from the chloroform fraction of A. trilobata leaves was only to assign that this plant has been already used and showed potential. Then, our study is to aggregate value to the knowledge about this plant and expand its use.

So, the choice for the essential oil as the object of this work do not include the study of the chloroform fraction, whose work is already reported in the literature, but includes the mono- and sesquiterpenes of the essential oil itself and its main compound, sulcatyl acetate.

-Why do you get the essential oil by hydrodistillation rather than chloroform extraction?

As mentioned previously, we did not work with the chloroform fraction. We used only the essential oil and the majoritary substance. In this work we use the hydrodistillation method.

Experimental Section

-Line 69. «Samples of Aristolochia trilobata…» Stems. Not samples.

The word was changed. Now it is in line 73.

-Line 69. Please indicate the geographic coordinates of the place of collection of A. trilobata stems.

Geographic coordinates: S 11° 14′ 22.4′′ and W 037° 25′ 00.5′′. it was included in methods section. Please see line 74

Results

-Line 197. Section 3.1. «Chemical analysis» repeats the information presented in your previous article «Volatile constituents of Aristolochia trilobata L. (Aristolochiaceae): A rich source of sulcatyl acetate» (http://dx.doi.org/10.5935/0100-4042.20140163). Why do you duplicate the data? Table 1 is identical to the table in the above article.

I also recommend you delete chemical information from the Experimental section 2.2 and 2.3 since these data were previously presented. 

We agree with Reviewer. We removed these parts

-Line 211. The phase designations should correspond to each other in the Experimental section and in the Results Section («first phase, second phase… 1st phase, 2nd phase».

We corrected along all the text.

-Line 228. Please check the link. I think the information from link 7 is presented.

In this line we had in the original version, the structure os SA. As asked by reviewer 1, we retired all information regarding chemical analyses, table of chemical components and methods of this part.

-Line 255. I do not see the data on the essential oil in the Figure 4. There is information only about sulcatyl acetate.

Our apoloizes. There was na error in the figure. Now we inserted the correct one. With results from EO and SA.

-Figure 4. Why L-NAME increases the effectiveness of sulcatyl acetate as an antinociceptive agent, although a decrease in activity would be expected, which is observed for other inhibitors (naloxone, atropine)?

L-NAME did not significantly increased antinociceptive effect of SA.

In our protocol none of antagonists significantly reversed the antinociceptive effects of EO or SA.

-Figure 5. I am confused by the description of the results in section 3.3.1. In my opinion according Figure 5, the best reduction in leucocyte number was observed for essential oil. Describe your results.

Reviewer is correct. We described the figure 5.

-Figure 5. Where is data on dexamethasone? How can you reason about the results of an experiment without data on the action of the reference substance?

Our apologies. It was our mistake. We inserted in the new version data concerning dexamethazone in all figures.

-Line 284. «As EO and SA presented significative effect reducing leukocyte migration…» Compared to what, the effect was significative?

We corrected the text. Please see line 282-286

-Figure 6, section 3.3.2. There is no reference substance. I cannot evaluate the data obtained.

Our apologies. It was our mistake. We inserted in the new version data concerning the positive control group (dexamethazone) in all figures.

-Figure 7, section 3.3.3. There is no reference substance. I cannot evaluate the data obtained.

Our apologies. It was our mistake. We inserted in the new version data concerning the positive control group (dexamethazone) in all figures.

-Figure 8, section 3.3.4. There is no reference substance. I cannot evaluate the data obtained.

Our apologies. It was our mistake. We inserted in the new version data concerning the positive control group (dexamethazone) in all figures.

-Figure 9, section 3.3.5. It makes no sense to discuss the data. There is no correct description of the location of groups of essential oil and sulcatyl acetate. No data for reference substance.

Our apologies. It was our mistake. We inserted in the new version data concerning the positive control group (L-NAME). We also inserted the indication of each graph

-Figure 10. What does it mean – «p-p38»? There is no reference substance. I cannot evaluate the data obtained. Why did you use a single dose in this experiment (30 µg/mL)? Before that you used 3 different concentrations.

We used only one dose (30 µg/mL) our intention was only to demonstrate a possible mechanism of action. As the higher dose did not affect enzymes expression. We did not showed lower doses.

We added a description for p-p38 MAPK. Please see lines 421-428.

Direct inhibitors of iNOS, such as L-NAME or L-NMMA, do not inhibit iNOS expression. They are only enzyme activity inhibitors.

To show a positive control group we added data obtained after preincubation of cells with dexamethasone. Please see figure 10 and lines 412-418

Discussion

-Line 369. «The present work demonstrated that the essential oil of A. trilobata and its majoritarian substance, sulcatyl acetate, present significant anti-inflammatory and antinociceptive effects». I can’t talk about the significance of the anti-inflammatory and antinociceptive effect without data on the effect of the reference compounds.

Our apolozise for our error. We did added all data regarding positive control groups, in all figures. We did not added any comparison in this first paragraph of discussion since our intention was to resume results obtained. But we added more discussion .

-Line 395. «We also demonstrated that EO and SA presented significant effect even when used by oral route a dose as lower as 1 mg/kg». I did not find in the manuscript data on the initial concentration less than 1 mg/kg.

Our apologize. We did a typographical error. The correct is: “...oral route a dose as LOW as 1 mg/kg...”. we corrected in the discussion section. Please see line 467.

-Line 409. «We also demonstrated that doses as lower as 1 mg/kg significantly reduced the licking response…». I did not find in the manuscript data on the initial concentration less than 1 mg/kg.

Our apologize. We did a typographical error. The correct is: “...oral route a dose as LOW as 1 mg/kg...”. we corrected in the discussion section. Please see lines 481.

References

-Please check links 7 and 10.

Both references were corrected

Round 2

Reviewer 1 Report

The authors have successfully addressed all major and minor issues.

Author Response

We would like to thank the Reviewer for the comments and improvement done in the Ms.

Reviewer 2 Report

Figure 5. Authors have added data on dexamethasone. However, compared to the same figure in the previous version of the manuscript, authors have changed the data on sulcacyl acetate (10 mg/g and 100 mg/g). What is the reason?

Figure 6. Where are the designations for this picture?

"We added a description for p-p38 MAPK. Please see lines 421-428". I see explanations only to p38, not phosphorilated p38 (p-p38). 

Author Response

1) Figure 5. Authors have added data on dexamethasone. However, compared to the same figure in the previous version of the manuscript, authors have changed the data on sulcacyl acetate (10 mg/g and 100 mg/g). What is the reason?

In the original version of the manuscript, the figure 5 was related to leukocyte migration into the pouch. And figure 6 was protein measurement.

As Reviewer asked to retire the chemical anayses, the figure 1 was retired. So, the old figure 5 in the new version is figure 4. And the old figure 6 (in the original version) is the figure 5 in the new version.

In summary:

                                            Original version             new version (submited in revision 1)

Leukocyte migration            figure 5                             figure 4

Protein measurement          figure 6                             figure 5

We did not change data concerning SA in figure 5. And we did not change the graph

We really don’t know what happned and we’d like to excuse our mistake

2) Figure 6. Where are the designations for this picture?

Figure 6 is NO production. The text and the reference indicating this figure are in lines 272 and 276.

3) "We added a description for p-p38 MAPK. Please see lines 421-428". I see explanations only to p38, not phosphorilated p38 (p-p38). 

We added more text better explaining about p-p38 (the phosphorylated and active formo f p38). Please see lines 409-411 and 416-422
